# Backdoor Attack with Imperceptible Input and Latent Modification

**Khoa Doan, Yingjie Lao, Ping Li**
Cognitive Computing Lab
Baidu Research
10900 NE 8th St. Bellevue, WA 98004, USA
{khoadoan106, laoyingjie, pingli98}@gmail.com

## Abstract

Recent studies have shown that deep neural networks (DNN) are vulnerable to various adversarial attacks. In particular, an adversary can inject a stealthy backdoor into a model such that the compromised model will behave normally without the presence of the trigger. Techniques for generating backdoor images that are visually imperceptible from clean images have also been developed recently, which further enhance the stealthiness of the backdoor attacks from the input space. Along with the development of attacks, defense against backdoor attacks is also evolving. Many existing countermeasures found that backdoor tends to leave tangible footprints in the latent or feature space, which can be utilized to mitigate backdoor attacks.

In this paper, we extend the concept of imperceptible backdoor from the input space to the latent representation, which significantly improves the effectiveness against the existing defense mechanisms, especially those relying on the distinguishability between clean inputs and backdoor inputs in latent space. In the proposed framework, the trigger function will learn to manipulate the input by injecting imperceptible input noise while matching the latent representations of the clean and manipulated inputs via a Wasserstein-based regularization of the corresponding empirical distributions. We formulate such an objective as a non-convex and constrained optimization problem and solve the problem with an efficient stochastic alternating optimization procedure. We name the proposed backdoor attack as Wasserstein Backdoor (WB), which achieves a high attack success rate while being stealthy from both the input and latent spaces, as tested in several benchmark datasets, including MNIST, CIFAR10, GTSRB, and TinyImagenet.

## 1 Introduction

In the past years, deep neural network (DNN) has successfully transformed many technological fields, such as object classification [26, 20], face recognition [31, 1], autonomous driving [53], security applications [19, 3], etc. Meanwhile, due to the underlying black-box nature, its security and privacy implications have also raised serious concerns recently. Efforts in the research community have exposed the vulnerability of DNN classifiers to various attacks [50, 41, 33]. For instance, adversarial examples leverage the difference between the classifier and human to misclassify specific inputs by adding imperceptible perturbations without altering the model [17]. Such attacks during the inference phase are categorized as evasion attacks [27, 5]. On the other hand, poisoning attacks attempt to inject malicious data points or manipulate the training process to either degrade the model accuracy [37, 45, 60] or cause misclassification for specific inputs (a.k.a. backdoor attacks) [8, 36, 34, 18].

In general, backdoor attacks aim at injecting a malicious behavior into a DNN model so that the model would perform normally on clean inputs but yield misclassification in the presence of the

35th Conference on Neural Information Processing Systems (NeurIPS 2021).

backdoor trigger (e.g., a specific pattern such as a small square [18]). Later on, many works adopt the concepts and techniques in adversarial examples to improve the stealthiness of the trigger against human observers [34, 2, 35]. Recent works have demonstrated more powerful backdoor attacks that are capable of mounting attacks with visual indistinguishable backdoor images [29, 55, 59, 39, 13]. For instance, WaNet [39] generates backdoor images with warping transformation to minimize input difference while LIRA [13] generates backdoor images with imperceptible conditional noise addition, resulting in much stealthier triggers.

To alleviate the threats originated from the ever-growing powerful backdoor attacks, several categories of countermeasures have been developed. One promising direction for backdoor detection entails identifying backdoor images by characterizing the distinguishable dissimilarity in the feature or latent representation between backdoor images and clean images [6, 54, 42, 47, 52]. These methods rely on the assumption that the injected backdoor would leave a noticeable fingerprint in the latent space. For example, activation clustering [6] and spectral signature [54] detect malicious samples by inspecting the clusters of the latent space and the spectrum of the covariance of latent representations, respectively. Thus, a stronger adaptive backdoor attack should also ensure its stealthiness from the latent space.

In this paper, we present a novel methodology for **a backdoor attack that is imperceptible from both the input and latent spaces**. We extend the concept of generating imperceptible backdoor triggers to the latent space by minimizing the Wasserstein distance between the latent representations of the clean and backdoor data, which significantly improves the effectiveness against the existing defense mechanisms, especially those aforementioned that rely on the distinguishability in latent space. We name the proposed method **W**asserstein **B**ackdoor, or **WB**. Our technical contributions are summarized below:

- We propose a non-convex, constrained optimization problem, which learns to poison the classifier with a backdoor whose trigger is visually imperceptible in the input space and whose poisoned samples have indistinguishable latent distribution to the latent distribution of the clean samples. The latent constraint is formulated via a variant of Wasserstein distance, called sliced-Wasserstein distance [24], between the two sets of clean and backdoor data.

- We then develop an efficient estimation of the sliced-Wasserstein distance by exploiting the discriminant directions of the trained classifier, instead of randomly sampling from the unit sphere. The proposed distance is a valid distance metric and requires significantly less computation, while yielding a better estimate than the existing calculations of the sliced-Wasserstein distance.

- Finally, we demonstrate the superior attack performance of the proposed method and its robustness against several representative defense mechanisms. Specifically, we show that the proposed method outperforms the state-of-the-art attacks in terms of latent indistinguishability, while maintaining similar attack success rates and input indistinguishability.

The rest of the paper is organized as follows. We review the background and related work in Section 2. In Section 3, we define the threat model. Section 4 presents the details of the proposed methodology. We evaluate the performance and compare to prior works in Section 5. Finally, Section 6 presents remarks and concludes this paper. We present more details about experimental settings and results as well as supporting proofs in the supplementary material.

## 2 Background and Related Work

### 2.1 Backdoor Attack

The increasing popularity of training outsourcing and machine learning as a service (MLaaS) has created potential security risks in the supply chain [10, 58]. One important security threat is backdoor attacks against DNNs, which have recently attracted a lot of attention. Backdoor attacks inject a malicious behavior by leveraging the redundancies inside the model such that the model responds to inputs with triggers maliciously (e.g., classify as a target class that would normally be considered as a wrong class by manual annotation), while preserving the benign behavior for clean inputs without the triggers. Hence, a typical backdoor embedding process is to train the model by minimizing the loss of the clean inputs and the corresponding labels as well as backdoor inputs (with triggers) and

the target class(es). A trigger is typically applied on a clean image by superimposing at a certain location (i.e., patch-based) [18, 34] or adding perturbations [44]. Various forms of the triggers have been investigated in the literature, including blended [8], sinusoidal strips (SIG) [2], reflection (ReFool) [35], and warping-based (WaNet) [39]. As we mentioned above, several techniques have been developed recently that can significantly reduce the visibility of the trigger in the input space to enhance the stealthiness of the backdoor attack [34, 2, 35]. In particular, WaNet uses a smooth warping field to generate backdoor images with unnoticeable modifications [39], while LIRA [13] alternates between the processes of trigger generation and backdoor injection to learn visually stealthy triggers. One prior work, Adversarial Embedding [51], also attempted to improve the latent indistinguishability of the backdoor attack by using adversarial regularization to minimize the distance between the latent distributions of the backdoor inputs and clean inputs.

## 2.2 Backdoor Defense

By exploring specific characteristics of the injected backdoor, various countermeasures have been proposed [6, 54, 16, 47, 9, 7, 42], although they are often circumvented by subsequent adaptive attacks. For instance, based on the property that a backdoor attack usually targets redundant weights or neurons based on the clean images, model pruning can be used to eliminate the injected backdoor [32]. In contrast, Neural Cleanse assumes a known subset of clean inputs to reverse-engineer possible trigger patches [56]. It is also possible to filter the images to nullify the presence of triggers at the test phase to defend against backdoor attacks [36, 30].

In this paper, we focus on optimizing the characteristics of backdoor attacks in the latent space. As we discussed above, the rationale behind this is that prior works have demonstrated backdoor images cause distinctive activations in the latent space from those of clean inputs. Hence, this distinguishable dissimilarity between clean images and backdoor images can be utilized for defense in both training [6, 54] and test phases [49, 23, 22]. Most of these approaches compute an outlier score to detect abnormal inputs that will be filtered afterward. For example, spectral signature [54] computes the outlier score based on the singular value decomposition of the covariance matrix of the latent representations, while CleaNN [22] leverages a concentration inequality to detect anomalous reconstruction errors that are then suppressed before the input entering the victim DNN.

This work proposes a method to minimize the difference between clean images and backdoor images in the latent space to improve the attack stealthiness. While doing this, we also optimize the visual imperceptibility in the input space, so that our proposed method can bypass visual inspection.

# 3  Threat Model

We consider the same threat model as in prior studies [44, 51, 39], which assumes the backdoor injection is performed at training and the adversary can access to the victim model including both structures and parameters. A successful backdoor attack over an image classification task should produce malicious behavior on images with the trigger, while otherwise working normally on clean images. However, in typical backdoor attacks, the poisoned images are visually inconsistent with natural images, which can be identified easily by human observers. Besides, these attacks usually leave a tangible trace in the latent space of the poisoned classifier; thus, some defense methods can easily detect and discard the poisoned models. To this end, we propose a stronger backdoor attack where the poisoned images are crafted with imperceptible perturbation in the input space to clean images as well as unnoticeable trace in the latent space. We advance the state-of-the-art by significantly enhancing the imperceptibility and robustness of the backdoor attack.

# 4  Proposed Methodology: Wasserstein Backdoor (WB)

## 4.1  Preliminaries

Consider the standard supervised classification task where one seeks to learn a mapping function $f_\theta : \mathcal{X} \longrightarrow \mathcal{C}$ where $\mathcal{X}$ is the input domain and $\mathcal{C}$ is the set of target classes. The task is to learn the parameters $\theta$ by using the training dataset $\mathcal{S} = \{(x_i, y_i) : x_i \in \mathcal{X}, y_i \in \mathcal{C}, i = 1, .., N\}$.

Following the standard training scheme of backdoor attacks, the classifier is trained with the combination of the clean and poisoned subsets of $S$. To create a poisoned sample, a clean training sample $(x, y)$ is transformed into a backdoor sample $(T(x), \eta(y))$, where $T$ is a backdoor injection function (also called the trigger function) and $\eta$ is the target label function. When training $f$ with the clean and poison samples, we alter the behavior of $f$ so that:

$$f(x) = y, \quad f(T(x)) = \eta(y), \tag{1}$$

for any pair of clean data $x \in \mathcal{X}$ and its corresponding label $y \in \mathcal{C}$. There are two commonly studied backdoor attack settings [18, 39, 51]: all-to-one and all-to-all. In the all-to-one attack, the label is changed to a constant target, i.e. $\eta(y) = c$; while for the all-to-all attack, the true label is one-shifted, i.e. $\eta(y) = (y + 1) \bmod |\mathcal{C}|$. In the existing works, the trigger function $T$ is usually selected before training $f$ and fixed during the training process of $f$.

## 4.2 Learning to Backdoor

Given the training dataset $\mathcal{S}$ and a loss function $\mathcal{L}$, e.g., cross entropy loss, empirical risk minimization can be used to learn the parameters $\theta$, as follows:

$$\theta^* = \arg\min_{\theta} \sum_{i=1}^{N} \mathcal{L}(f_{\theta}(x_i), y_i).$$

The goal of this work is to learn a trigger function $T_{\xi} : \mathcal{X} \longrightarrow \mathcal{X}$ and a classification model $f_{\theta}$ in such a way that the clean image $x$ and its corresponding backdoor image $T(x)$ are visually consistent in the input space while the backdoor attack does not leave a detectable trace in the latent space of the poisoned classifier. When $f$ is a neural network, $\phi(x)$ can be the output of an intermediate, hidden layer of $f$, which captures some high-level abstractions of the input. Note that we require the classifier to perform normally on the clean sample, $x$, compared to the classifier's vanilla version, but change its prediction on the poisoned image, $T(x)$, to the target class $\eta(y)$.

To generate a trigger and poison the image, we follow the prior work [13] and formulate the trigger function as a conditional noise generator $g$, as follows:

$$T_{\xi}(x) = x + g_{\xi}(x), \quad ||g_{\xi}(x)||_{\infty} \leq \epsilon \quad \forall x \tag{2}$$

The generator function $g_{\xi}$ takes an input $x$ and generates an artificially imperceptible noise on the same input space, which guarantees the stealthiness of the backdoor attack. We can design such generator function as an autoencoder or the more complex U-Net architecture [43].

With the above objectives and notations, similar to [13, 4], we can formulate the task into the following constrained optimization problem:

$$\min_{\theta} \sum_{i=1}^{N} \alpha \mathcal{L}(f_{\theta}(x_i), y_i) + \beta \mathcal{L}(f_{\theta}(T_{\xi^*(\theta)}(x_i)), \eta(y_i)) \tag{3}$$

$$s.t. \quad \xi^* = \arg\min_{\xi} \sum_{i=1}^{N} \mathcal{L}(f_{\theta}(T_{\xi}(x_i)), \eta(y_i)) + \mathcal{R}_{\phi}(\mathcal{F}_c, \mathcal{F}_b)$$

where $\mathcal{R}_{\phi}$ is the regularization constraint of the clean and poisoned representations, denoted as $\mathcal{F}_c = \{\phi(x_i) : i = 1, .., N\}$ and $\mathcal{F}_b\{\phi(T(x_i)) : i = 1, .., N\}$, respectively.

In this problem, a learned classification model with a specific parameter configuration $\theta$ is associated with an optimal yet stealthy backdoor trigger function, which is trained to poison the model. The classifier is trained to minimize a linear combination of clean and targeted backdoor objectives. The parameters $\alpha$ and $\beta$ control the mixing strengths of the clean and backdoor loss signals. The trigger function is trained to perturb an image within its $\ell_{\infty}$ ball in the input space, so that the loss towards the attack target class is minimized while regularizing the latent representations of the backdoor images.

## 4.3 Stealthy Latent Representation via Wasserstein Regularization

In practical applications, latent-space defense methods investigate the abnormal trace of incoming data points with respect to the previous stream of data. These traces exist primarily because of the

fact that the clean and backdoor latent representations are separated or distributed differently (e.g., the separated clusters of the clean and poison representations that can be seen in Figures 2 and 3). Thus, we aim to minimize such distributional difference through the regularization constraint $\mathcal{R}_\phi$. Since we cannot assume that the two latent distributions have common support or their density functions are known, commonly-used divergences, such as $f$-divergences [40, 15] (which include KL and JSD), are difficult to minimize. Instead, we consider the Wasserstein-2 distance and formulate the regularization constraint as follows:

$$\mathcal{R}\phi(\mu, \nu) = \left( \inf_{\gamma \in \Pi(\mu,\nu)} \int_{(x,z) \sim \gamma} p(x, z)||x - z||_2 dx dz \right)^{1/2} \quad (4)$$

where $\mu$ and $\nu$ are marginal probability measures defined by empirical samples $\mathcal{F}_c$ and $\mathcal{F}_b$ of the latent representations of the clean and poisoned data, respectively.

Estimating the Wasserstein distance also has some challenges. From the primal domain, computing the infimum in Equation (4) is particularly difficult since the data distributions are not fixed or known. On the other hand, employing the Kantorovich-Rubinstein duality requires a separate, parameterized Lipschitz function and a minimax solver, which increases the complexity of the proposed problem. Fortunately, for one-dimensional continuous measures, the Wasserstein distance has an elegant yet closed-form solution. Let $q_\mu$ and $q_\nu$ be the corresponding density functions of $\mu$ and $\nu$, respectively. The Wasserstein-2 distance between one-dimensional measures $\mu$ and $\nu$ can be given by:

$$\mathcal{W}(\mu, \nu) = \left( \int_0^1 ||(F_\mu^{-1}(z) - F_\nu^{-1}(z)||_2 dz \right)^{1/2} \quad (5)$$

where $F_\mu(z) = \int_\infty^z q_\mu(\rho) d\rho$ and $F_\nu(z) = \int_\infty^z q_\nu(\rho) d\rho$ are the cumulative distribution functions. Inspired by the efficiency of this solution and its successful applications in a variety of tasks [12, 24, 14], we propose to first find a family of one-dimensional representations, e.g., through the linear projections, and approximate the Wasserstein distance as a function of these one-dimensional marginals, as follows:

$$\mathcal{R}_\phi(\mathcal{F}_c, \mathcal{F}_b) \approx \left( \frac{1}{L} \sum_{l=1}^{L} [\mathcal{W}(\mathcal{F}_c^{\theta_l}, \mathcal{F}_b^{\theta_l})]^2 \right)^{1/2} \quad (6)$$

where $\mathcal{F}_c^{\theta_l} = \{\theta_l^T \phi(x_i) : i = 1, .., N\}$ and $\mathcal{F}_b^{\theta_l} = \{\theta_l^T \phi(T(x_i)) : i = 1, .., N\}$ contains the projections of the clean and poisoned datasets into a one-dimensional direction defined by $\theta_l$ (a slice). Typically, $\theta_l$ is drawn from a uniform distribution on the unit sphere. This formulation is also known as the sliced-Wasserstein distance (SWD) [12, 24]. One particular problem with this approach is that the random nature of the slices could lead to several non-informative directions; i.e., the sliced distances are close to 0 in directions that do not lie on the manifolds of the data. Consequently, a large number $L$ of random directions are needed to approximate the sliced-Wasserstein distance, which increases the computational complexity of the estimation.

To remedy this issue, we avoid the uniform sampling of the unit sphere and select directions that contain discriminant information of the two data sources, by exploiting the following fact in the classification task. For backdoor samples of a target class $c_1 \in \mathcal{C}$, created from clean samples of some other class $c_2 \in \mathcal{C}$, the projections into an output dimension represent meaningful discriminant information that distinguishes the backdoor samples (from class $c_2$) and the clean samples (from class $c_1$). Thus, we propose to replace the uniform linear projections of SWD with the projections into the output layer. When the latent space is the penultimate layer of the classifier, such projections are equivalent to the following approximation:

$$\mathcal{R}_\phi(\mathcal{F}_c, \mathcal{F}_b) \approx \left( \frac{1}{|\mathcal{C}|} \sum_{c=1}^{|\mathcal{C}|} \left[ \mathcal{W}(\mathcal{F}_c^{W_{c,:}}, \mathcal{F}_c^{W_{c,:}}) \right]^2 \right)^{1/2}. \quad (7)$$

where $W_{c,:}$ is a row of the matrix $W \in \mathbb{R}^{|\mathcal{C}| \times d}$ ($d$ is the dimension of the latent space), which is the normalized parameter matrix between the penultimate and the output layers.

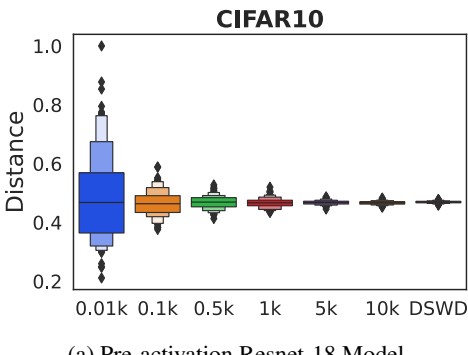
(a) Pre-activation Resnet-18 Model

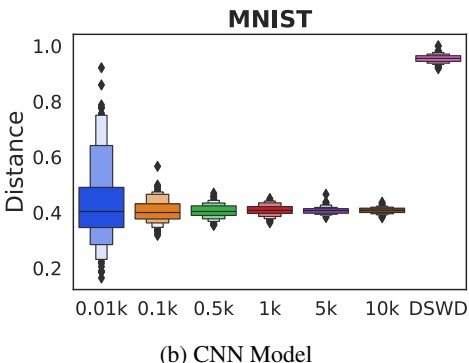
(b) CNN Model

Figure 1: Distance estimates (normalized) in the latent space for SWD with different number of sampled directions (between 10 to 10,000) and DSWD.

Empirically, Figure 1 shows the estimated SWD with different numbers of random directions and the proposed calculation, so called DSWD, when the latent space is defined at the penultimate layer of the classifier. The dimension of the latent space is 512 for both MNIST and CIFAR10 datasets. Each distance is computed on a random sample of 1000 clean and 1000 backdoor images, and each calculation is repeated 100 times. It can be seen that with only a fraction of slices, DSWD achieves a significantly smaller variance than that of the SWD estimates. Furthermore, in MNIST, the selected directions of DSWD leads to higher distance estimates than SWD, which means that DSWD selects more discriminant directions than SWD while SWD underestimates the distance between the two empirical samples. In addition, we show that DSWD is a valid distance metric of the latent distributions. The detailed proof is presented in the supplementary material.

**Theorem 1.** *When the latent space is the penultimate layer of a neural network, the proposed DSWD distance is a valid distance function of probability measures in this space.*

**Remark 1.** *Since existing defense methods choose the penultimate layer of a neural network. as the space to perform the defense analysis, in most cases, we can employ the proposed DSWD calculation.*

**Remark 2.** *To preserve the clean classification performance, the classifier seeks optimal parameters that lead to similar predictions of clean samples from the same class. The goal of the trigger function is to make the poisoned samples classified toward a different class. This leads to an adversarial game between the classifier and the trigger functions.*

DSWD also has a significantly better computational efficiency than SWD. In most problems, SWD requires a large number of random directions, typically between 1000 to 10,000, in order to provide a reliable estimate of the distance [38, 14]. In DSWD, the number of random directions is fixed to the number of possible output labels, which is typically small for many classification problems.

### 4.4 Optimization

The non-convex, constrained optimization in Equation (3) is challenging because of its non-linear constraint. In general, we can alternately update one of $f$ and $T$ while keeping the other fixed, similar to training GANs. However, it is difficult and slow for the classifier to reach an acceptable performance on the clean data, i.e., similar accuracy to that of the vanilla classifier.

Under the alternating update scheme, we observe that on MNIST, the poisoned classifier can reach the acceptable clean-data performance after several epochs; while on other more complex datasets (i.e. CIFAR10, GTSRB, and TinyImagenet), this procedure results in sub-optimal clean-data performance. One possible explanation is that training the vanilla classifier with complex architecture and dataset to reach a decent accuracy is already a difficult and time-consuming task (e.g., 2 to 3 epochs to reach the optimal performance on MNIST but several hundreds of epochs on the other datasets).

Fortunately, we observe that after training the classifier and the trigger functions in an alternating update scheme for a certain number of epochs (denoted as Stage I), we can fix the trigger function and only train the classifier for the remaining epochs (denoted as Stage II). This two-stage training scheme is adopted in our experiments.

# 5 Experimental Results

## 5.1 Experimental Setup

We demonstrate the effectiveness of the proposed method through a range of experiments on four widely-used datasets for backdoor attack study: **MNIST**, **CIFAR10**, **GTSRB** and **TinyImagenet**. For these experiments, we follow the previous works [51, 54, 6, 39] and select the penultimate layers of the classifiers as the latent space for the defense experiments. The implementation of WB was based on the PaddlePaddle deep learning platform.

**Architectures:** For the classifier $f$, we consider several popular models: Pre-activation Resnet-18 [20], VGG [46], DenseNet [21] for CIFAR10 and GTSRB datasets, and Resnet-18 for TinyImagenet. For the MNIST dataset, we employ a CNN model.

**Hyperparameters:** For the baselines, we train the classifiers using the SGD optimizer with an initial learning rate of 0.01 and a learning rate decay of 0.1 after every 100 epochs. For other hyperparameters, we follow the proposed setup in [39] for all datasets. We use the same configurations for WB. We train the classifier and trigger functions alternately (Stage I) for 10 and 50 epochs for MNIST and the other datasets, respectively, and fine-tune the classifier (Stage II) for another 40 epochs and 450 epochs for MNIST and the other datasets, respectively. To achieve a high-degree stealthiness of WB, we pick $\epsilon$ as small as 0.01 for all datasets. In general, the larger the value of $\epsilon$, the easier the trigger functions can be learned and the more successful the attacks are.

## 5.2 Attack Performance

We present the attack success rates of the proposed WB method, along with a comparison to two state-of-the-art methods, i.e., WaNet [39] and LIRA [13]. Both LIRA and Wanet's attack performances are significantly better than other approaches, including BadNets [18], and are two of the strongest existing methods that generate very stealthy triggers on the images. We first poison the classifier using the backdoor attack methods in both all-to-one and all-to-all settings and record the performance of the classifier on both clean and backdoor test samples. For all-to-one, we randomly pick the target label $\hat{c}$ (i.e., $\eta(y) = \hat{c} \ \forall y$), while for all-to-all, the target label function is defined as $\eta(y) = (y + 1) \ mod \ |\mathcal{C}| \ \forall y$, which is widely used to evaluate the backdoor-related works [39, 18, 6, 13]. Note that this all-to-all attack setting is more challenging than the all-to-one setting, especially on datasets with a large number of classes such as TinyImagenet.

The classification accuracy on the clean test samples and the attack success rate for each method is represented in Table 1 and Table 2 for the all-to-one and all-to-all settings, respectively. As we can observe from these tables, all the methods can achieve high clean-data accuracies and attack success rates. While WB's attack performance slightly drops compared to LIRA's performance, WB is significantly more stealthy in the latent space, as being discussed next.

Table 1: Attack Performance: All-to-one Attack

| Dataset | WaNet | | LIRA | | WB | |
|---------|-------|--------|------|--------|------|--------|
| | Clean | Attack | Clean | Attack | Clean | Attack |
| MNIST | 0.99 | 0.99 | 0.99 | 1.00 | 0.99 | 0.99 |
| CIFAR10 | 0.94 | 0.99 | 0.94 | 1.00 | 0.94 | 0.99 |
| GTSRB | 0.99 | 0.98 | 0.99 | 1.00 | 0.99 | 0.99 |
| TinyImagenet | 0.57 | 0.99 | 0.58 | 1.00 | 0.57 | 0.99 |

Table 2: Attack Performance: All-to-all Attack

| Dataset | WaNet | | LIRA | | WB | |
|---------|-------|--------|------|--------|------|--------|
| | Clean | Attack | Clean | Attack | Clean | Attack |
| MNIST | 0.99 | 0.95 | 0.99 | 0.99 | 0.99 | 0.96 |
| CIFAR10 | 0.94 | 0.93 | 0.94 | 0.94 | 0.94 | 0.94 |
| GTSRB | 0.99 | 0.98 | 0.99 | 1.00 | 0.99 | 0.98 |
| TinyImagenet | 0.58 | 0.58 | 0.58 | 0.59 | 0.58 | 0.58 |

## 5.3 Latent-Space Defense

Recent works on backdoor defense have found that backdoor attacks tend to leave a tangible trace in the latent space of the poisoned classifier. Activation Clustering [6] and Spectral Signature [54] are two representative defenses used for analyzing the latent space in prior work [51]. In this section, we also examine the latent space of the backdoor-injected classifiers through the lens of these defense methods.

### 5.3.1 Learned Latent Representation and Activation Clustering

It has been shown in [6] that in a poisoned classifier, the latent representations of the clean and backdoor samples form separate clusters, which can be easily detected using clustering methods such as K-means. The authors also recommend a process called exclusionary reclassification to determine which cluster is poisoned and re-train the poisoned classifier.

In Figure 2 and Figure 3, we can observe highly separated clusters (for samples with the sample predictions of $y = 0$) in the latent space when we omit the latent regularization term $\mathcal{R}_\phi$ in WB (Baseline), which is similar to LIRA [13]. However, when $\mathcal{R}_\phi$ is included, the latent representations of the clean and backdoor samples are distributed similarly. Without well-separated clusters of the clean and poisoned samples, the exclusionary reclassification process in the activation clustering is not effective against the attacks.

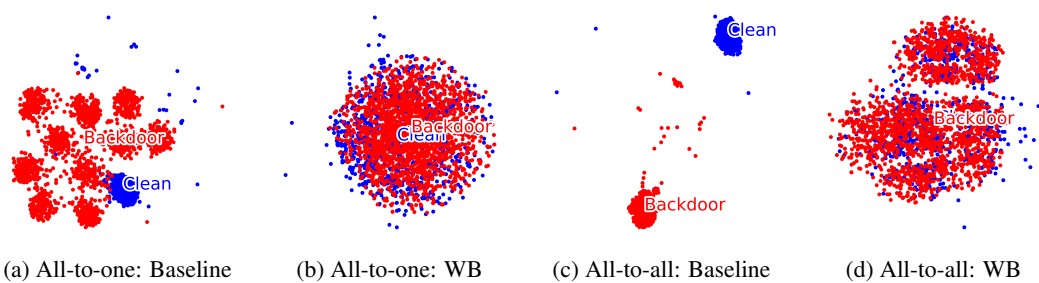

| (a) All-to-one: Baseline | (b) All-to-one: WB | (c) All-to-all: Baseline | (d) All-to-all: WB |

Figure 2: MNIST: t-SNE embedding in the latent space. Baseline is WB without $\mathcal{R}_\phi$.

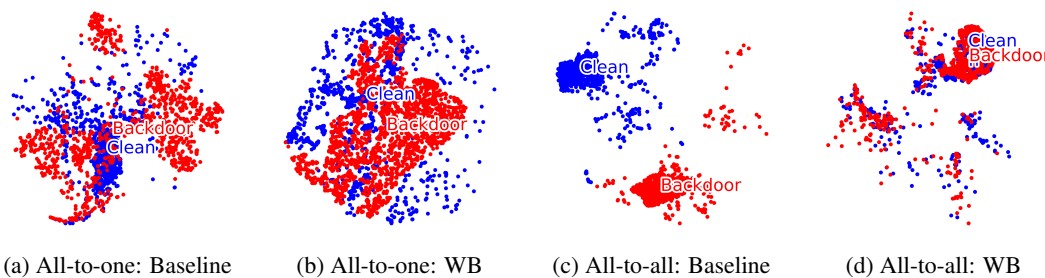

| (a) All-to-one: Baseline | (b) All-to-one: WB | (c) All-to-all: Baseline | (d) All-to-all: WB |

Figure 3: CIFAR10: t-SNE embedding in the latent space. Baseline is WB without $\mathcal{R}_\phi$.

Quantitatively, we present the quality scores (i.e., the adjusted Rand Index) of the clustering step in Table 3. The adjusted Rand Index is 1 when the samples form two distinct clusters and is close to 0 for a random separation. We compare WB with BadNets [18] and Adversarial Embedding [51], which is the state-of-the-art backdoor attack method with stealthy latent space. As we can observe in this table, the defense is most successful on BadNets since there exists a perfect clustering of the

Table 3: Adjusted Rand Index in All-to-one Attack

| Model | Dataset | Rand Index (BadNets) | Adversarial Embedding | | WB | |
|---|---|---|---|---|---|---|
| | | | Rand Index | Attack | Rand Index | Attack |
| DenseNet | CIFAR10 | 0.979 | 0.1820 | 0.764 | 0.0382 | 0.998 |
| DenseNet | GTSRB | 0.997 | 0.2710 | 0.914 | 0.0135 | 0.997 |
| VGG | CIFAR10 | 0.998 | 0.0006 | 0.962 | 0.0002 | 0.999 |
| VGG | GTSRB | 0.997 | 0.6420 | 0.743 | 0.1010 | 0.999 |

clean and poisoned samples (Rand Index ≥ 0.95). While Adversarial Embedding is more resistant against the defense, WB is significantly more stealthy against the defense since the values of Rand Index are all very close to 0. Note that, similar to BadNets, WaNet also does not pass this defense (please see the supplementary material).

### 5.3.2 Spectral Signature Defense

The work in [54] proposes a defense method that identifies and removes backdoor samples using the Spectral Signature. For data from each predicted class, Spectral Signature first finds the top singular value of the covariance matrix of the latent vectors of the data. Then it computes the correlation score to this singular value for each sample and those samples with the outlier scores are flagged as backdoor samples. While Spectral Signature is a sample filtering-based defense method, the inspection of the correlation scores can also be useful to verify whether there is a tangible trace in the latent space of the classifier.

Following the same experiments in [54], we first pick 5,000 clean samples and 500 backdoor samples for each dataset. Then, we plot the histograms of the correlation scores for both sets of samples. As we can observe in Figure 4, there is no clear separation between the scores of the backdoor samples and those of the clean samples.

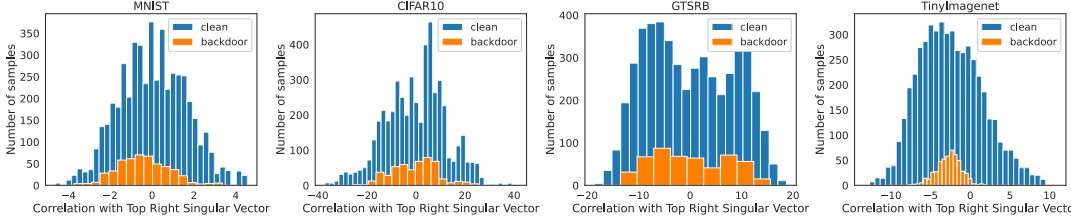

Figure 4: Defense experiments of the all-to-one attack against Spectral Signature. The correlations of the clean and backdoor samples with the top singular vector of the covariance matrix *in the latent space are not separable*.

### 5.4 Model Mitigation Defense

In this section, we evaluate the robustness of WB against another popular defense, Neural Cleanse [56], which is model mitigation defense based on a pattern optimization approach. Specifically, Neural Cleanse searches for the optimal patch pattern for each possible target label that induces a misclassification to that label. It then quantifies whether any of the optimal backdoor trigger pattern is an outlier via a metric called Anomaly Index. The model has a backdoor if the Anomaly Index is greater than 2 for any class. The anomaly indices are presented in Figure 5.

It can be seen that both WaNet and WB can pass the detection of Neural Cleanse, similar to that of the vanilla classifier (Clean). In MNIST and CIFAR10, WB even achieves smaller Anomaly Indices than

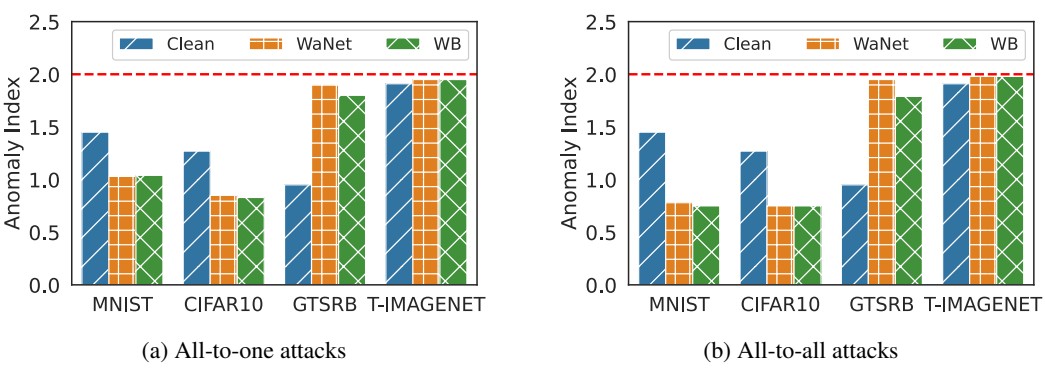

(a) All-to-one attacks          (b) All-to-all attacks

Figure 5: Backdoor attacks against Neural Cleanse defense.

those of the vanilla models. Note that popular backdoor attacks, such as BadNets, can be defended by Neural Cleanse in most of these datasets [56].

## 5.5 Input Perturbation Defense

In this section, we study the stealthiness of WB against STRIP [16], a representative detection based backdoor defense mechanism. Given the classifier and an input image, STRIP first perturbs the image and determines the presence of a backdoor in the model according to the entropy of the predictions of these perturbed images (i.e., if the predictions are consistent or not).

In Figure 6, we plot the entropy of clean and backdoor images, which are computed by STRIP. We can observe that the distribution of entropy of the backdoor samples is similar to that of the clean samples. In other words, STRIP fails to detect backdoor samples generated by WB, which further validates the advantage of the proposed method.

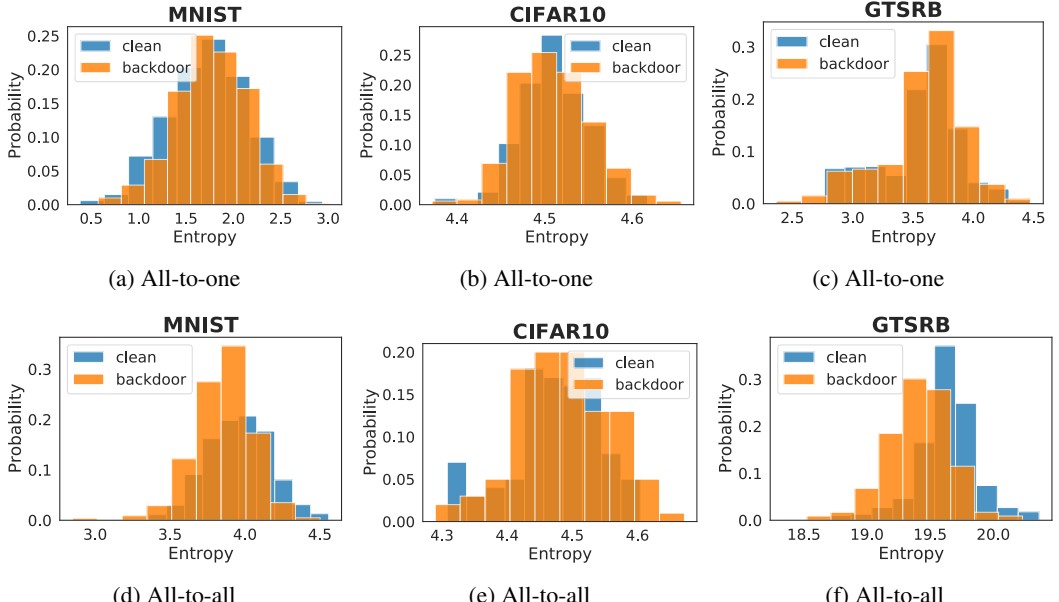

Figure 6: Performance against STRIP defense.

Additional experiments for demonstrating the stealthiness of WB against several other defense approaches can be found in the supplementary material.

## 6   Conclusion

This paper presented a novel methodology for a backdoor attack that is imperceptible from both the input and latent spaces, i.e., Wasserstein Backdoor (WB). WB learns a trigger function that adds visually imperceptible noise to an input image and minimizes the distributional difference via a novel sliced Wasserstein distance formulation between representations of the clean and backdoor images in the latent space of the trained classifier. We comprehensively evaluated the performance of the proposed method on various image classification benchmark models over a wide range of datasets. Our experimental results demonstrated that the proposed method could significantly improve the effectiveness against the existing defense mechanisms, especially those relying on the distinguishability in latent space.

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
