# Backdoor Attack with Imperceptible Input and Latent Modification

**Khoa Doan, Yingjie Lao, Ping Li**
Cognitive Computing Lab
Baidu Research
10900 NE 8th St. Bellevue, WA 98004, USA
{khoadoan106, laoyingjie, pingli98}@gmail.com

**Supplementary Materials.** This document provides additional details, analysis, and experimental results. We begin by discussing the detailed experimental setup and implementation of the methods in Section A. Then, we provide additional empirical experiments against several other defense methods in Section B, and a discussion on the stealthiness of the backdoor images in the input space in Section C. Finally, we provide the supporting proofs for the claims in the main paper in Section D.

## A   Detailed Experimental Setup

### A.1   Datasets

As we described in the main paper, we use four datasets, MNIST, CIFAR10, GTSRB, and TinyImagenet, to evaluate our method. Note that MNIST, CIFAR10, and GTSRB have been widely used in the literature of backdoor attacks on DNN. On the other hand, the use of a more complex dataset, TinyImagenet, enables better evaluation for multiple-target backdoor attacks such as all-to-all, thanks to the diversity of images in TinyImagenet and its large number of classes.

- **MNIST** [28] is a subset of the larger dataset available from the National Institute of Technology. This dataset consists of 70,000 grayscale, $28 \times 28$ images, divided into a training set of 60,000 images and a test set of 10,000 images. We applied random cropping and random rotation as data augmentation for the training process. During the evaluation stage, no augmentation is applied. Link to the dataset: http://yann.lecun.com/exdb/mnist

- **CIFAR-10** is first introduced by [25]. It is a labeled subset of the 80-millions-tiny-images dataset, collected by Alex Krizhevsky, Vinod Nair, and Geoffrey Hinton, consists of 60,000 color images at the resolution of $32 \times 32$, out of which 10,000 images are randomly selected as the query set, and the remaining images used as the retrieval set. Link to the dataset: https://www.cs.toronto.edu/~kriz/cifar.html

- **GTSRB** (German Traffic Sign Recognition Benchmark [48]) is used as an official dataset for the challenge held at the International Joint Conference on Neural Network (IJCNN) 2011. GTSRB consists of 60,000 images, divided into 43 classes, with resolutions varying from $32 \times 32$ to $250 \times 250$. The training set contains 39,209 images, while the test set has 12,630. In our experiments, GTSRB input images are all resized into $32 \times 32$ pixels, then applied random crop and random rotation in training. In the evaluation stage, no augmentation is used. Link to the dataset: http://benchmark.ini.rub.de/?section=gtsrb&subsection=dataset

- **TinyImagenet** is a smaller subset of the large-scale Imagenet dataset [11], which is introduced in [57]. This dataset consists of 200 image classes. The training set has 500 images per class, resulting in 100,000 images, while the test set has 50 images per class, resulting in 10,000 images. TinyImagenet input images are all resized into $64 \times 64$ resolution. Random crop and random rotation are applied in the training stage. No augmentation is used in the evaluation stage. Link to the dataset: http://cs231n.stanford.edu/tiny-imagenet-200.zip

## A.2 Noise Generator Models

For MNIST, we use a self-defined autoencoder, which is detailed in Table 4. For the other datasets, we employ the UNet architecture [43]. We observe only a slight performance difference between the simpler autoencoder and the complex UNet on these datasets.

Table 4: Autoencoder-based generator network used in this paper.

| Layer | Filters | Filter Size | Stride | Padding | Activation |
|---|---|---|---|---|---|
| Conv2D | 16 | $3 \times 3$ | 3 | 1 | BatchNorm2D+ReLU |
| MaxPool2d | - | $2 \times 2$ | 2 | 0 | - |
| Conv2D | 64 | $3 \times 3$ | 2 | 1 | BatchNorm2D+ReLU |
| MaxPool2d | - | $2 \times 2$ | 2 | 0 | - |
| ConvTranspose2D | 128 | $3 \times 3$ | 2 | - | BatchNorm2D+ReLU |
| ConvTranspose2D | 64 | $5 \times 5$ | 3 | 1 | BatchNorm2D+ReLU |
| ConvTranspose2D | 1 | $2 \times 2$ | 2 | 1 | BatchNorm2D+Tanh |

## A.3 Models

In this work, we use a simple CNN classifier for MNIST, which was also used in WaNet [39]. For convenience, we include the detailed architecture in Table 5. For CIFAR10 and GTSRB datasets, we use PreActResnet18 [20]. For TinyImagenet, we use Resnet18 [20].

Table 5: CNN model architecture for MNIST.

| Layer | Filters | Filter Size | Stride | Padding | Activation |
|---|---|---|---|---|---|
| Conv2D | 32 | $3 \times 3$ | 2 | 1 | ReLU |
| Conv2D | 64 | $3 \times 3$ | 2 | 0 | ReLU |
| Conv2D | 64 | $3 \times 3$ | 2 | 0 | ReLU |
| Linear | 512 | - | - | - | ReLU |
| Conv2D | 10 | - | - | - | Softmax |

## A.4 Training Hyperparameters

Table 6 provides additional details to Section 5.1 in the main paper.

Table 6: Experimental setup and parameters for the datasets we used in this paper.

| | MNIST | CIFAR10 | GTSRB | TinyImagenet |
|---|---|---|---|---|
| Optimizer | SGD | SGD | SGD | SGD |
| Batch Size | 128 | 128 | 128 | 128 |
| Learning Rate | 0.01 | 0.01 | 0.01 | 0.01 |
| Learning Rate Schedule | 10,20,30,40 | 100,200,300,400 | 100,200,300,400 | 100,200,300,400 |
| Learning Rate Decay | 0.1 | 0.1 | 0.1 | 0.1 |
| Training Epochs | 50 epochs | 1000 epochs | 1000 epochs | 1000 epochs |
| **WB Only** | | | | |
| $\epsilon$ | 0.01 | 0.01 | 0.01 | 0.01 |
| $\alpha$ | 0.5 | 0.5 | 0.5 | 0.5 |
| $\beta$ | 0.5 | 0.5 | 0.5 | 0.5 |
| Stage I | 50 epochs | 50 epochs | 50 epochs | 50 epochs |
| $T$'s Optimizer | SGD | SGD | SGD | SGD |
| $T$'s Learning Rate | 0.001 | 0.001 | 0.001 | 0.001 |
| Clean Accuracy | 0.99 | 0.94 | 0.99 | 0.57 |

In our experiments, choosing $0.5$ for both $\alpha$ and $\beta$ allows us to achieve clean-data accuracy similar to that of the vanilla classifier and state-of-the-art attack success rates (almost 100% in most of the experiments).

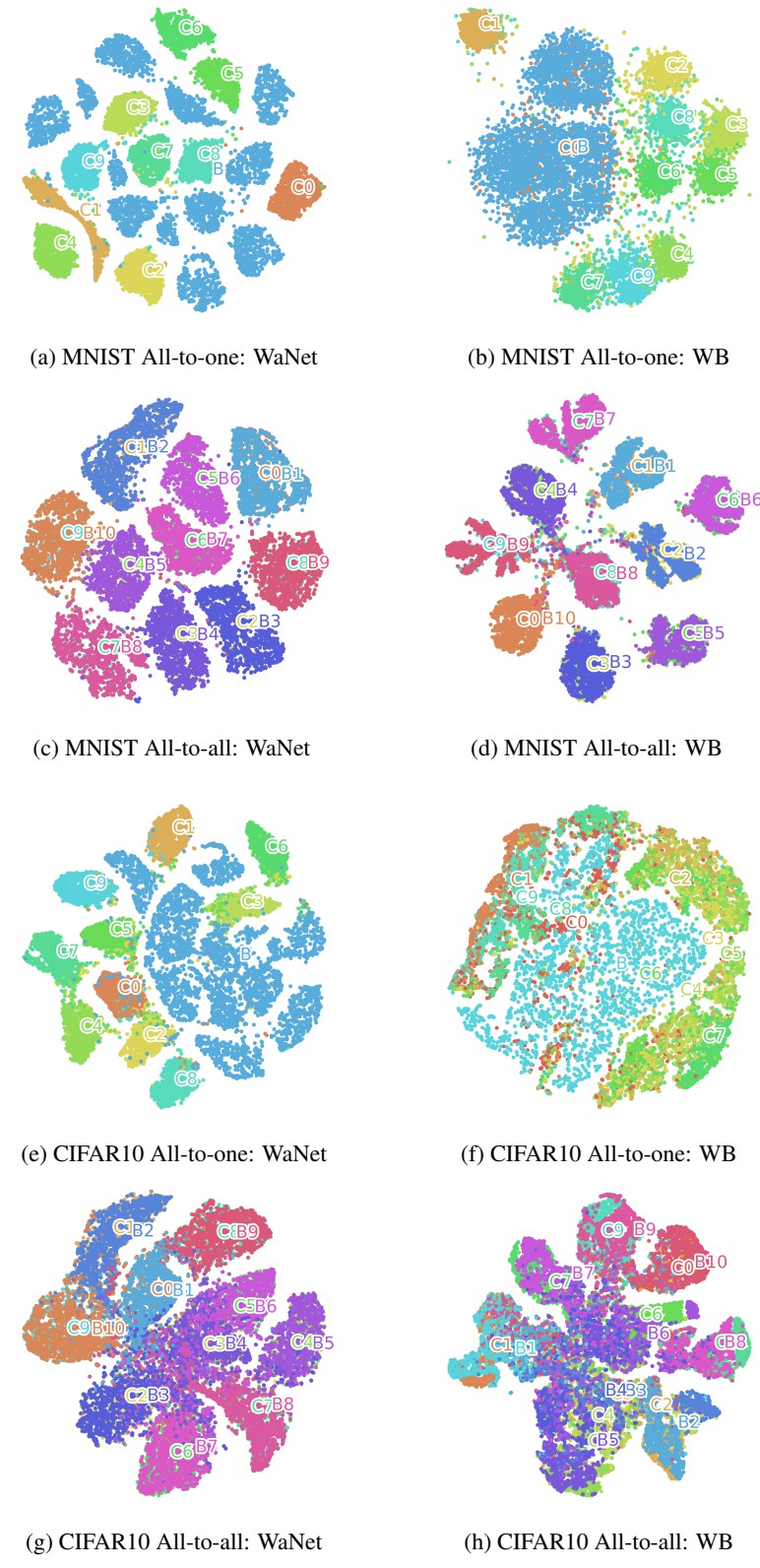

(a) MNIST All-to-one: WaNet

(b) MNIST All-to-one: WB

(c) MNIST All-to-all: WaNet

(d) MNIST All-to-all: WB

(e) CIFAR10 All-to-one: WaNet

(f) CIFAR10 All-to-one: WB

(g) CIFAR10 All-to-all: WaNet

(h) CIFAR10 All-to-all: WB

Figure 7: Characteristics of latent space. **Ci**: Clean samples with predicted label **i**. **Bi**: Backdoor samples with predicted label **i**. Note that the numerical labels represent the encoded labels from categorical classes.

# B  Additional Defense Experiments

## B.1  Backdoor Attack Performance

The standard deviations of the attack performance for WB are as follows: 0.0007 for MNIST, 0.0004 for CIFAR10, 0.0004 for GTSRB, 0.0007 for TinyImagenet. It can be seen that the standard deviations are very small. We also compare to the performance reported in the Adversarial Embedding paper here: 0.90 for CIFAR10 (significantly worse than 0.99 of both WaNet and WB), and 0.94 for GTSRB (again, worse than 0.98 of WaNet and 0.99 of WB).

Table 7: Normalized Wasserstein distance of clean and poisoned samples for a predicted class.

| Dataset | Attack Setting | Clean | WaNet | LIRA | WB |
|---------|----------------|-------|-------|------|-----|
| MNIST | all-to-one | $0.010 \pm 0.002$ | $0.351 \pm 0.002$ | $0.325 \pm 0.009$ | $0.045 \pm 0.005$ |
| MNIST | all-to-all | $0.006 \pm 0.001$ | $0.718 \pm 0.004$ | $0.840 \pm 0.002$ | $0.021 \pm 0.002$ |
| CIFAR10 | all-to-one | $0.027 \pm 0.006$ | $0.253 \pm 0.004$ | $0.261 \pm 0.003$ | $0.084 \pm 0.003$ |
| CIFAR10 | all-to-all | $0.027 \pm 0.006$ | $0.471 \pm 0.003$ | $0.463 \pm 0.003$ | $0.125 \pm 0.004$ |

Clean: distance between two random subsets of the clean data.
WaNet/LIRA/WB: distance between poisoned and clean samples for each attack method.

## B.2  Latent Space Characteristics

In Section 5.3.1 of the main paper, we provide a visualization of the latent space of the clean and backdoor poisoned samples of a specific predicted class. In this section, we provide an additional inspection of the latent space. Specifically, Figure 7 shows the visualization of the latent space for all data points, while Table 7 presents the Wasserstein distances between the clean and poisoned samples from different classes that quantify the visual observation.

As we can observe from Figure 7, WB encourages poisoned samples to be indistinguishably closer to the clean samples. In all-to-one cases, the poisoned samples are closer to the clean samples of the target class (i.e., 'C0'), while in the all-to-all cases, poisoned samples of different classes separately become closer to the clean samples of those classes. In contrast, for the compared baseline, WaNet, poisoned samples of a class are completely separated from the clean samples of that class. Correspondingly, as we can observe in Table 7, the Wasserstein distances between the poisoned samples and the clean samples are significantly smaller than the distances of the similar experiments for the WaNet and LIRA.

## B.3  Performance against Fine-pruning Defense

Fine-pruning [32] is a network analysis based defense method. Given a set of neurons of the neural network classifier, it analyzes their activations on a set of clean images and detects the dormant neurons, assuming they are more likely to tie to the backdoor. The dormant neurons are then gradually pruned to mitigate the backdoor.

In Figure 8, we evaluate WB against fine-pruning by plotting the accuracies on the clean and backdoor data when different numbers of the dormant neurons are pruned. It can be seen that at no point does the backdoor accuracy drop considerably more significant than the clean accuracy. This suggests that the backdoor mitigation defense of fine-pruning is also ineffective against WB.

# C  Visual Inspection Experiments

In this section, we study the stealthiness of the triggers generated by different backdoor attack methods. As we can observe from Figure 9, the perturbation-based attacks (BadNets [18], Blended [8], SIG [2], and ReFool [35]) can be easily detected through human inspection because of their noticeable visual trigger patterns. In WaNet [39], while the triggers are less perceivable, we still find a considerable amount of "difficult" cases where WaNet's attacks can fail under human inspection. In contrast, the images generated by our proposed method appear more natural and genuine.

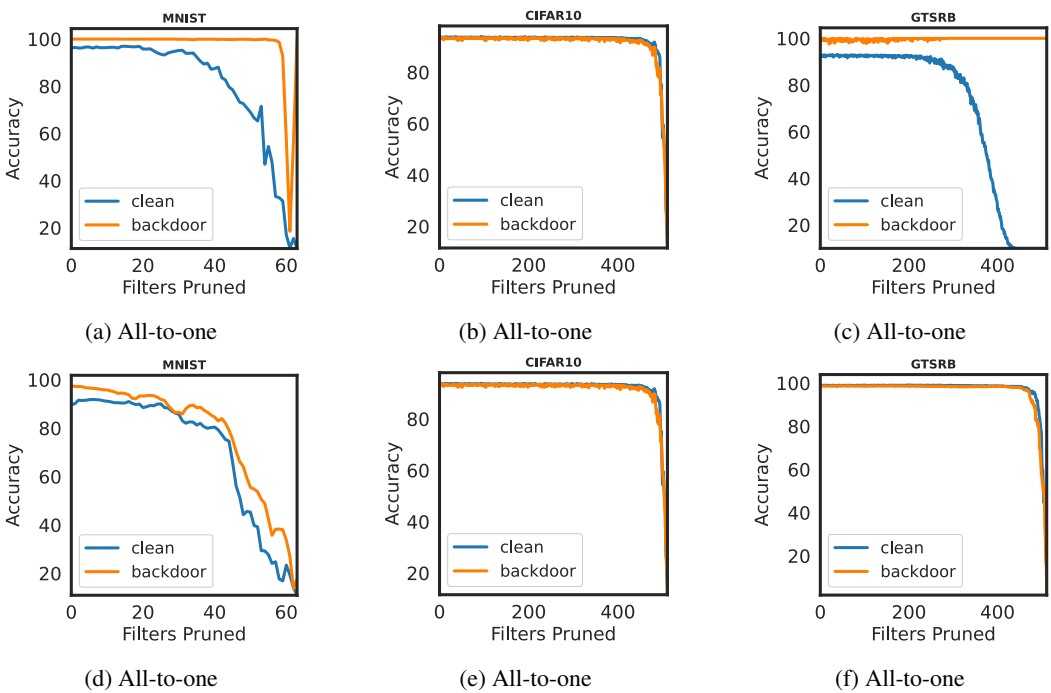

Figure 8: Performance against fine-pruning defense.

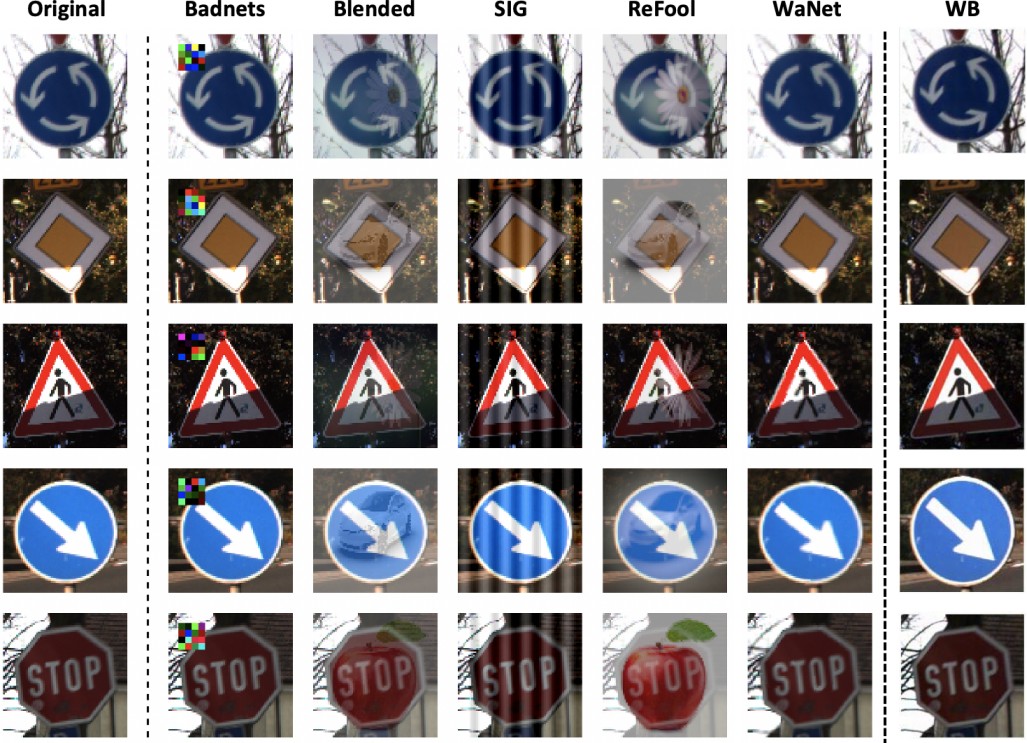

Figure 9: Backdoor images created from different backdoor methods. In BadNets and similar perturbation-based methods, the trigger patterns are visible, which makes the attack very easy to be detected. In WaNet and WB, the trigger patterns are more difficult to be detected. However, in WaNet, edges from common shapes such as a circle, rhombus, or triangle are deformed (e.g., the circle is not entirely round, or edges from rhombus or rectangles are not straight), thus the backdoor can be detected with closer inspection.

## D   Proof of Theorem 1

**Theorem 1.** *When the latent space is the penultimate layer of a neural network, the proposed DSWD distance is a valid distance function of probability measures in this space.*

*Proof.* We first prove that DSWD satisfies the triangle inequality. Let $\mathcal{F}_1$, $\mathcal{F}_2$, and $\mathcal{F}_3$ be empirical samples. Then, since the projections (i.e., $\mathcal{F}_k^{\theta_l} = \{\theta_l^T x : x \in \mathcal{F}_k\}$ $\forall k$) are linear, we have:

$$
\begin{aligned}
\mathcal{R}_\phi(\mathcal{F}_1, \mathcal{F}_3) &= \left( \frac{1}{|\mathcal{C}|} \sum_{c=1}^{|\mathcal{C}|} \left[ \mathcal{W}(\mathcal{F}_1^{W_{c,:}}, \mathcal{F}_3^{W_{c,:}}) \right]^2 \right)^{1/2} \\
&\leq \left( \frac{1}{|\mathcal{C}|} \sum_{c=1}^{|\mathcal{C}|} \left[ \mathcal{W}(\mathcal{F}_1^{W_{c,:}}, \mathcal{F}_2^{W_{c,:}}) + \mathcal{W}(\mathcal{F}_2^{W_{c,:}}, \mathcal{F}_3^{W_{c,:}}) \right]^2 \right)^{1/2} \\
&\leq \left( \frac{1}{|\mathcal{C}|} \sum_{c=1}^{|\mathcal{C}|} \mathcal{W}(\mathcal{F}_1^{W_{c,:}}, \mathcal{F}_2^{W_{c,:}})^2 \right)^{1/2} + \left( \frac{1}{|\mathcal{C}|} \sum_{c=1}^{|\mathcal{C}|} \mathcal{W}(\mathcal{F}_2^{W_{c,:}}, \mathcal{F}_3^{W_{c,:}})^2 \right)^{1/2} \\
&= \mathcal{R}_\phi(\mathcal{F}_1, \mathcal{F}_2) + \mathcal{R}_\phi(\mathcal{F}_2, \mathcal{F}_3)
\end{aligned}
$$

where the first inequality is because $\mathcal{F}$ is a metric, and the second inequality follows from the application of Minkowski inequality.

Since $\mathcal{W}(\mathcal{F}^{W_{c,:}}, \mathcal{F}^{W_{c,:}}) = 0$ for any $c$, it is trivial to see that $\mathcal{R}_\phi(\mathcal{F}, \mathcal{F}) = 0$. For the reverse direction of identity of discernibility, the proof can be laid out as follow: $\mathcal{R}(\mathcal{F}_1, \mathcal{F}_2) = 0$ is equivalent to $\mathcal{W}(\mathcal{F}_c^{W_{c,:}}, \mathcal{F}_b^{W_{c,:}}) = 0$ for all $c$. Since it can be shown that $\mathcal{F}_b^{W_{c,:}}$ and $\mathcal{F}_c^{W_{c,:}}$ are injective, this implies that $\mathcal{F}_1 = \mathcal{F}_2$.

Finally, since $\mathcal{W}(.,.)$ is a non-negative and symmetric, $\mathcal{R}_\phi(.,.)$ is also non-negative and symmetric.

Therefore, DSWD is a valid distance.

$\square$