# OpenReview forum: "Backdoor Attack with Imperceptible Input and Latent Modification"
_NeurIPS.cc/2021/Conference — NeurIPS 2021 Poster_

### Official Review · Reviewer_mknK · 2021-06-27

**Rating:** 4
**Confidence:** 3

**Summary:**

This paper proposes a backdoor attack approach for image classifiers. The generated backdoor examples are imperceptible in both input space and the latent space. To do this, the paper optimize the noise to matching the latent representations of the clean and manipulated inputs via a Wasserstein-based regularization of the corresponding empirical distributions. The effectiveness of the method is evaluated on MNIST, CIFAR10, GTSRB, and TinyImagenet, and bypassing spectral signature defense and model mitigation defense.

**Limitations And Societal Impact:**

Yes.

**Main Review:**

Pros:
1. The paper is overall clear written.

2. The paper measures performance on four datasets MNIST, CIFAR10, GTSRB, and TinyImagenet.

3. It is good to see the attack is measured on two defense method and bypass them.

Cons:
1. Marginal Gain. The paper mainly compares with WaNet, and from Table 2, the performance is almost the same. For example, for GTSRB and TinyImageNet, the number is exactly the same. For the MNIST and CIFAR-10, WB is only 0.01 better than WaNet. As MNIST and CIFAR-10 are small datasets, where baselines are already very high, the author should consider using more complex dataset such as ImageNet. On the other hand, on the larger Tiny ImageNet dataset, the gain is 0. Why the method performs worse on All-to-all attack than all-to-one attack?

2. In Figure 5, it seem both the baseline WaNet and the proposed can pass the detection. What is the special advantage of the proposed method compared with WaNet then?

3. Given that WB's performance is close to WaNet in numbers. What is the motivation of this paper? What is the challenge the addressed by this paper but not the others?

4. Figure 2,3, the author should show t-sne for WaNet too.



**Time Spent Reviewing:**

2

---

> ### Author Response · Authors · 2021-08-10
> **We sincerely thank the reviewer for the thoughtful comments.**
>
> **Q1: Marginal Gain. The paper mainly compares with WaNet, and from Table 2, the performance is almost the same. For example, for GTSRB and TinyImageNet, the number is exactly the same. For the MNIST and CIFAR-10, WB is only 0.01 better than WaNet. As MNIST and CIFAR-10 are small datasets, where baselines are already very high, the author should consider using more complex dataset such as ImageNet. On the other hand, on the larger Tiny ImageNet dataset, the gain is 0. Why the method performs worse on All-to-all attack than all-to-one attack?**
>
> R1: Thanks for your comments. Please note that the main contribution of this paper is to extend the concept of imperceptible backdoor from the input space to the latent representation, which significantly improves the effectiveness against the existing defense mechanisms (as we described in L9-11 in the Abstract). The very recent work WaNet (ICLR'21) achieved much better input-space imperceptibility than prior works, which is also the reason we consider WaNet as state-of-the-art in this direction of research and compare our performance to it. Compared to WaNet, WB achieves as good as, if not better, than WaNet from the aspects of attack success rate  (Tables 1 and 2)  and stealthiness at the input space (Table 8 in the supplementary material). However, from the latent space, WB is much more stealthy than prior works such that our method can bypass the representative defenses used for analyzing the latent space, as we demonstrated in Section 5.3.
>
> The reason that we evaluated these datasets, including the larger TinyImageNet, is to follow the similar setups and network architectures as in previous backdoor attack and defense papers for a fair comparison to the earlier methods.
>
> The objective of an all-to-one attack is to manipulate all the images in the dataset to one target label. In contrast, an all-to-all attack attempts to change the labels of images at the class level (i.e., images from different original classes will have different target labels).  One common setup for all-to-all attacks is to define the target label as one-shifted (e.g., for MNIST, the target label for digit "0" with the backdoor is 1, the target label for digit "1" with the backdoor is 2, the target label for digit "2" with the backdoor is 3, etc.), which is used in our experiments as described in Section 5.2. Thus, compared to all-to-one attacks that only have a single target label, all-to-all attacks involve multiple target labels and each target label associates to less number of poisoned images correspondingly, making such all-to-all attacks more challenging. The trend where the attack performance is worse on all-to-all attacks than all-to-one attacks is also consistent with the existing works, including BadNets and WaNet.
>
> **Q2: In Figure 5, it seem both the baseline WaNet and the proposed can pass the detection. What is the special advantage of the proposed method compared with WaNet then?**
>
> R2: Thanks for pointing this out. Figure 5 illustrates the performance against Neural Cleanse, which is a model-mitigation defense based on a pattern optimization approach at the input space as opposed to analyzing the latent space. Since our objective is mainly to improve the stealthiness at the latent space while achieving similar performance from the aspects of attack success rate and stealthiness at the input space, we expect that WB and WaNet exhibit similar performance against Neural Cleanse. As we mentioned in the response above, from the latent space, WB is much more stealthy than prior works such that our method can bypass the representative defenses used for analyzing the latent space, as we demonstrated in Section 5.3.
>
> **Q3: Given that WB's performance is close to WaNet in numbers. What is the motivation of this paper? What is the challenge addressed by this paper but not the others?**
>
> R3: As we noted above, the main contribution of this paper is to extend the concept of imperceptible backdoor from the input space to the latent representation, which significantly improves the effectiveness against the existing defense mechanisms.
>
> **Q4: Figure 2,3, the author should show t-sne for WaNet too.**
>
> R4: Please kindly find the comparison of t-SNE between WaNet and WB in Figure 6 in the supplementary material. It can be seen from the results that WB encourages poisoned samples to be indistinguishably closer to the clean samples, while, in WaNet, poisoned samples of a class are separated from the clean samples of that class. Correspondingly, as we can observe in Table 7, the Wasserstein distances between the poisoned samples and the clean samples are significantly smaller than the distances of the similar experiments for the WaNet.

---

### Official Review · Reviewer_mJMq · 2021-07-16

**Rating:** 7
**Confidence:** 4

**Summary:**

This paper proposes a novel backdoor attack that is stealthy in both the input and latent spaces. It argues that the previously proposed backdoor techniques leave tangible footprints in the latent space, thus being easily detected. The authors propose to remove that weakness by adding a regularization term that minimizes the Wasserstein distance between the clean and backdoor embeddings when joint training the trigger function and the classification network. The complex Wasserstein distance is approximated with the sliced-Wasserstein distance (SWD). The authors consider the latent representation at the penultimate layer, thus further simplify SWD computation. The proposed backdoor attack has a high clean and attack accuracy while being stealthy under both input/latent space inspection and backdoor defenses, verified on various benchmark datasets.

**Limitations And Societal Impact:**

## Limitations
* The proposed method only protects the latent representations at the penultimate layer, which is a bit disappointed. I wonder if removing the backdoor footprint in the penultimate-layer features could eliminate the footprints in other-layer features. If not, how to extend the proposed method to the other layers?
* The cost function in Equation 3 only regulates the penultimate-layer features (Fc, Fb) but not the inputs (x, T(x)). How can the proposed method achieve imperceptible backdoor in the input space, as shown in the Supplementary PDF?
* In Equation 7, is the projection of a penultimate-layer feature via a row of the normalized parameter matrix of the last layer the corresponding logit? If yes, this equation can be further simplified.
* The mismatching between SWD and DSWD in Fig. 1b is concerning. The discussion in L212 seems to be wrong. DSWD is computed with only one direction while SWD is averaged over many directions; thus, SWD should be more accurate and lower.
* Minor comments:
- L286: The improvement, if exists, is very subtle. I would say two methods have similar performance.
- L309: Better move that result from the  appendix to Table 3.
- L109: "... can **pass** visual inspection".

## Societal Impacts
I agree with the discussion at the end of the paper. This paper helps increase awareness of potential backdoor attacks and facilitates the further development of secure and trustworthy DNN models and powerful defensive solutions.

**Main Review:**

* Originality: The proposed method is new. It addresses a problem that is not considered in the previous backdoor attacks. It designs imperceptible backdoor attacks both in latent and input space. A simplified version of the sliced-Wasserstein distance is employed for that purpose. It cites a good amount of related works.

* Quality: The submission is mostly technically sound. The proposed backdoor attack has a high clean and attack accuracy while being stealthy under both input/latent space inspection and backdoor defenses, verified on various benchmark datasets. The experiments were done thoroughly to support the claims. It is a complete piece of work.

* Clarity: The submission is well written and organized.

* Significance: This work is significant to the security research community.

**Time Spent Reviewing:**

12

---

> ### Author Response · Authors · 2021-08-10
> **Thank you the reviewer for the thorough reading of our paper and valuable suggestions.**
>
> **Q1: The proposed method only protects the latent representations at the penultimate layer, which is a bit disappointed. I wonder if removing the backdoor footprint in the penultimate-layer features could eliminate the footprints in other-layer features. If not, how to extend the proposed method to the other layers?**
>
> R1: Our attack method is designed to be aligned with the latent-space detection methods, including both Spectral Signature and Activation Clustering, which inspect the latent representations at the penultimate layer. These prior works show that backdoor attacks usually leave a tangible trace at the penultimate layer that can be detected. We think that looking at other-layer features and reducing the footprint on all the layers is an interesting future work, requiring an independent study of the effectiveness for both the existing defense methods and backdoor attacks.
>
> **Q2: The cost function in Equation 3 only regulates the penultimate-layer features (Fc, Fb) but not the inputs (x, T(x)). How can the proposed method achieve imperceptible backdoor in the input space, as shown in the Supplementary PDF?**
>
> R2: The imperceptibility in the input space is achieved via the backdoor injection function with a conditional noise generator (Equation (2)), which adds artificially imperceptible noise (as the trigger) to the image. The magnitude of this noise is controlled by the parameter $\epsilon$ (please see the values used in our experiments in Table 6 in the supplementary material).
>
> **Q3: In Equation 7, is the projection of a penultimate-layer feature via a row of the normalized parameter matrix of the last layer the corresponding logit? If yes, this equation can be further simplified.**
>
> R3: Yes, the output is the normalized logits. The formulation in this equation (Equation (7)) is intended for an easier comparison to Equation (6) and to have an emphasis on the improvement (smaller and fixed number of projections) of the proposed DSWD calculation (Equation (7))  over SWD (Equation (6)). However, we'd like to thank the reviewer for the simplification suggestion, which we will incorporate in the later revision of the paper.
>
> **Q4: The mismatching between SWD and DSWD in Fig. 1b is concerning. The discussion in L212 seems to be wrong. DSWD is computed with only one direction while SWD is averaged over many directions; thus, SWD should be more accurate and lower.**
>
> R4: Thank you for this observation. Interestingly, this is actually one of the limitations of SWD, which involves random directions with non-informative separations of the two distributions. This problem is even worse in higher dimensional space due to the curse of dimensionality. This limitation results in expensive computational complexity and higher variance (when fewer directions are used, as shown in this figure), which makes SGD training less efficient. On the other hand, DSWD is significantly more efficient at finding a smaller set of most informative directions. Thus, the estimated distance, which is the average of the 1-D Wasserstein distances along with these directions, is expected to be higher. Its variance is also shown to be lower. In this way, DSWD's objective is similar to Max Sliced Wasserstein Distance (MSWD) [20], but MSWD requires a separate network to estimate the distance. In fact, the number of directions in DSWD is equivalent to the number of classes as opposed to only one. The reported distances in Figure 1 are the average of the 1-D Wasserstein distances along with the projected directions, which is the numbers in the x-axis (i.e., 10, 100, 500, 1000, 5000, 10000) for SWD and 10 (since there are 10 classes in CIFAR10 and MNIST) for DSWD.
>
> **Q5: Minor comments:
> L286: I would say two methods have similar performance.
> L309: Better move that result from the appendix to Table 3.
> L109: "... can pass visual inspection".**
>
> R5: Thanks for pointing out these minor errors and suggestions. We will correct these wordings and move the results accordingly in the later version.

---

> > ### Comment · Reviewer_mJMq · 2021-08-12
> > **More question on imperceptible nackdoor in the input space**
> >
> > Thanks for your answers. They addressed most of my concerns.
> >
> > As for Q2, I see that you use a small noise perturbation as backdoor. I checked the supplementary and found $\epsilon$ = 0.005. In [0-255] value range, it is only ~ 1.275. Is this perturbation too small? Also, your backdoor images in Figure 9 are smoother than other images (even the clean ones). Did you apply any image smoothing or interpolation?

---

> > > ### Author Response · Authors · 2021-08-13
> > > **Thank you the reviewer for the response and the additional comment.**
> > >
> > > **Q6: I checked the supplementary and found $\epsilon = 0.005$. In [0-255] value range, it is only ~ 1.275. Is this perturbation too small?**
> > >
> > > R6: Thanks for your comment. Our objective is to have a smaller $\epsilon$ to achieve better stealthiness. We empirically tried a range of $\epsilon$ values and found this value to be small for the stealthiness while also sufficiently large for a successful attack.
> > >
> > > **Q7: Your backdoor images in Figure 9 are smoother than other images (even the clean ones). Did you apply any image smoothing or interpolation?**
> > >
> > > R7: Thanks for the observation. We did not apply any image smoothing or interpolation. The trigger injection mechanism in our attack is generated directly from a conditional generator, and we did not perform any post-processing. Thus, the smoothing effect is not expected. We've also verified the same clean and WB's backdoor images from the original samples used for Figure 9, but the observed smoothness does not exist. We believe this smoothness is probably caused by some accidental effects during the process of creating Figure 9 in the PDF version.

---

> > > > ### Comment · Reviewer_mJMq · 2021-08-15
> > > > **Updated your score. Please correct Figure 9**
> > > >
> > > > I have updated your score to 7.
> > > > Please update Figure 9 to show your actually backdoor images in the revised version.

---

### Official Review · Reviewer_KsEU · 2021-07-17

**Rating:** 7
**Confidence:** 4

**Summary:**

This paper considers the backdoor attack settings, where the adversary has full control over the training process and tries to encode a trigger function into the classifier such that the classifier fails at test time when the trigger function is activated. They propose a new algorithm that trains a parameterized trigger function at the same as the classifier in a min-max fashion. To ensure that the trigger function is not detected, the paper add a regularization term that ensures that the latent representation of the modified samples and the clean samples have the same distribution, they propose to use a sliced wasserstein distance to do this.

**Limitations And Societal Impact:**

Yes

**Main Review:**

**Originality**:

The idea of parametrizing the trigger function as a generative model is new in the context of backdoor attacks. However, using an autoencoder to learn an attack patterns is not entirely new and has been proposed in the context for example of the No-Box threat model, I think this should be mentioned and consider as related work. The most related work also using a min-max formulation is (Bose et al. 2020).

Another issue with the related work is L111-113 which are slightly incorrect. [47] doesn't assume they have access to the parameters and [7] assume they have no knowledge and access to the model.

The rest of the related work is well documented, and the contributions are put into the context of existing work on backdoor attacks and defense.

**Quality**:

Overall the claims are well supported, I only have two minor comments:
1. Theorem 1 is slightly incorrect, in the appendix you only prove that the proposed distance is a pseudometric. You also need to prove $\mathcal{R}_\phi(\mathcal{F}_1, \mathcal{F}_2) = 0 \Rightarrow mathcal{F}_1 = mathcal{F}_2$, in order to prove that it is a proper distance function.
2. It would be nice to have confidence interval for the experiments. In particular for Table 1 and 2, right now the difference between WaNet and WB is pretty small, and it's hard to make any significant claim without the confidence interval. I'm also wondering why you don't compare to other attack strategies in this table, in particular against Adversarial Embedding which also seems to be effective against latent-space defense.

I really enjoyed the Figure 1, that clearly shows the efficiency of DSWD. I'm not very surprised by this and think the regularizer act similarly to an entropy regularizer that prevent the attacker from being "too good" but instead force the attacker to be just good enough to fool the classifier.

**Clarity**:
The paper is really well written and the contributions are nicely explained. I was not familiar with backdoor attacks so this part was difficult to understand at first, but once familiar the rest of the paper was very clear.

**Significance**:
The proposed approach seems like a viable approach to generate backdoor attacks that are hard to detect with current defense techniques. I believe this will force people to look into more robust defense against backdoor attacks.

**Additional references**:
- Bose et al. "Adversarial Example Games." ( NeurIPS 2020).

**Time Spent Reviewing:**

4

---

> ### Author Response · Authors · 2021-08-10
> **We thank you the reviewer for the positive reviews and valuable comments.**
>
> **Q1: Need to prove $R_{\phi}(F_1, F_2) \Rightarrow F_1 = F_2$. Is DSWD a metric (as mentioned in Theorem 1) or pseudometric?**
>
> R1: Thank you for the comment. DSWD is a metric. For the reverse direction of identity of discernibility, since the linear projection into the output layer is injective, following [20], it can be similarly shown that if $R_{\phi}(F_1, F_2)$ is 0, then $F_1 = F_2$. We will make this part clearer in the later version.
>
>
> **Q2: It would be nice to have confidence interval for the experiments. In particular for Table 1 and 2, right now the difference between WaNet and WB is pretty small, and it's hard to make any significant claim without the confidence interval. I'm also wondering why you don't compare to other attack strategies in this table, in particular against Adversarial Embedding which also seems to be effective against latent-space defense.**
>
> R2: Thanks for the suggestions. The attack performance of WB is consistent across different random initializations. The standard deviations of the attack performance for WB are as follows: 0.0007 for MNIST, 0.0004 for CIFAR10, 0.0004 for GTSRB, 0.0007 for TinyImagenet. It can be seen that the standard deviations are very small.
>
> Please note that the main contribution of this paper is to extend the concept of imperceptible backdoor from the input space to the latent representation, which significantly improves the effectiveness against the existing defense mechanisms. The very recent work WaNet (ICLR'21) achieved much better imperceptibility than prior works, which is also the reason we consider WaNet as state-of-the-art in this direction of research and compare our performance to it. We show that WB is much more stealthy than prior works such that our method can bypass the representative defenses used for analyzing the latent space. Nonetheless, we also compare to the performance reported in the Adversarial Embedding paper here: 0.90 for CIFAR10 (significantly worse than 0.99 of both WaNet and WB), and 0.94 for GTSRB (again, worse than 0.98 of WaNet and 0.99 of WB). These discussions will be included in the later version.
>
> **Q3: I really enjoyed the Figure 1, that clearly shows the efficiency of DSWD. I'm not very surprised by this and think the regularizer act similarly to an entropy regularizer that prevent the attacker from being "too good" but instead force the attacker to be just good enough to fool the classifier.**
>
> R3: Thank you for this interesting analogy. It is indeed the case that the attacker has to balance between being "too aggressive" and "stealthy in latent space" to achieve a better overall performance. This balancing act (or regularization), however, is effective as WB's attacks can be stealthy in both input and latent spaces without sacrificing the attack success rates (compared to state-of-the-art backdoor attack methods).
>
> **Q4: However, using an autoencoder to learn an attack patterns is not entirely new and has been proposed in the context for example of the No-Box threat model, I think this should be mentioned and consider as related work. The most related work also using a min-max formulation is (Bose et al. 2020).**
>
> R4: Thanks for pointing out this interesting work, which is designed for adversarial examples (as opposed to backdoor attacks). We will include this reference and add discussions about the implications of techniques from adversarial examples on backdoor attacks in the later version.

---

> > ### Comment · Reviewer_KsEU · 2021-08-30
> > **Thanks for the clarifications**
> >
> > Thanks for the clarifications you should mention [20] in your proof.

---

> > > ### Author Response · Authors · 2021-08-31
> > > **Thank you for your suggestion.**
> > >
> > > Thank you! We will include this reference in the proof for the later version of the manuscript.

---

### Official Review · Reviewer_TF3P · 2021-07-17

**Rating:** 7
**Confidence:** 3

**Summary:**

This paper proposes how to improve the stealthiness of backdoor attacks. To explain, the distribution of latent vectors has been an effective method to defend against backdoor attacks. However, this paper claims that an adversary can intentionally make backdoor attacks sneakier so that adding backdoor noise does not change the latent vector distribution too much. As the result, the authors propose a new attack method, Wasserstein Backdoor (WB), that has an imperceptibly small trigger and shows small differences in the latent vector distribution.

**Ethical Concerns:**

I do NOT think that this paper violated the NeurIPS ethics guidelines.

**Limitations And Societal Impact:**

As most attack-related papers do, this paper contains research on attacks that may have negative societal impacts. However, such societal impacts are discussed well by the authors in the conclusion.

**Main Review:**

Originality: The authors detailed enough about their contribution to improving existing backdoor attacks. This paper seems to be the first work that tried to generate an imperceptible backdoor attack from both the input and latent space.

Quality: This is a very well-written paper, containing both formal discussions and experimental supports.

Theoretically, the paper mostly focuses on how to estimate the regularization constraints efficiently, using the sliced-Wasserstein distance (SWD). The problem of this approach is again pointed out and the remedy is proposed.

Experimentally, the authors first evaluate WB with an existing method with a comparison to an existing method (WaNet). Then, the desired property (WB should generate a backdoor that is not detected by defenses utilizing latent space) is verified against two different defenses: Spectral Signature and Neural Cleanse. In my opinion, the paper could be improved further by comparing its attack performance to other backdoor attacks. For example, does the additional constraint (that the generated backdoor should not change the latent representations from that of the clean input) degrade the attack performance compared to the backdoor attack without such constraint? If so, how much is the performance degradation? Is there some tradeoff between attack performance and recognizability in the latent space?

Clarity: The writing is very clear and understandable. I suggest the authors check all notations being correct once again (e.g., subscript in equation 4).

Significance: The topic (backdoor attack) is an important field of research 1. to understand the machine learning behavior in an adversarial setting, 2. to construct safer real-world ML applications against potential threats. The fact that a backdoor can be imperceptible even from the latent space throws more challenges to the research community working on defenses.


**Time Spent Reviewing:**

5-6 hours

---

> ### Author Response · Authors · 2021-08-10
> **Thank you the reviewer for the positive reviews and insightful comments.**
>
> **Q1: Does the additional constraint (that the generated backdoor should not change the latent representations from that of the clean input) degrade the attack performance compared to the backdoor attack without such constraint? If so, how much is the performance degradation? Is there some tradeoff between attack performance and recognizability in the latent space?**
>
> R1: Compared to backdoor attack without the constraint at the latent space, such as in the case of WaNet, WB achieves as good as, if not better, attack success rate (Tables 1 and 2 in the paper) and stealthiness at the input space (Table 8 in the supplementary material). However, from the latent space, WB is much more stealthy than prior works since our method can bypass the representative defenses used for analyzing the latent space (as demonstrated in Section 5.3). In conclusion, WB is able to achieve the latent space stealthiness without sacrificing the attack success rates by regularizing the latent space of the poisoned classifier so that the learned representations of the backdoor and clean samples are indistinguishable.

---

### Decision · Program_Chairs · 2021-09-27

**Decision:**

Accept (Poster)

**Comment:**

This paper extends the notion of stealthy backdoors to the latent space (the authors focus on the penultimate layer). The manuscript shows how the proposed attack is able to evade prior defenses for backdoors. Optimizing an objective in the latent space is not completely novel to the adversarial ML area - it has been done in adversarial examples research (e.g., https://arxiv.org/abs/1511.05122) - but it is here demonstrated in the context of backdoors attacks which creates its own sets of challenges. Thus, there is merit to the work proposed here. In particular, experiments show how the proposed attack is able to obtain similar performance to prior approaches while also improving the stealthiness in latent space. I encourage the authors to take into account discussions from the reviews while preparing the camera ready of their manuscript, and to open-source their code.